# Intention to Be Vaccinated for COVID-19 among Italian Nurses during the Pandemic

**DOI:** 10.3390/vaccines9050500

**Published:** 2021-05-12

**Authors:** Marco Trabucco Aurilio, Francesco Saverio Mennini, Simone Gazzillo, Laura Massini, Matteo Bolcato, Alessandro Feola, Cristiana Ferrari, Luca Coppeta

**Affiliations:** 1Department of Medicine and Health Sciences “V. Tiberio”, University of Molise, 86100 Campobasso, Italy; marco.trabuccoaurilio@unimol.it (M.T.A.); laura.massini@unimol.it (L.M.); 2EEHTA-CEIS, DEF Department, Faculty of Economics, University of Rome “Tor Vergata”, 00133 Rome, Italy; mennini@uniroma2.it (F.S.M.); simone.gazzillo@uniroma2.it (S.G.); 3Institute for Leadership and Management in Health, Kingston University, London KT1 2EE, UK; 4Legal Medicine, University of Padua, Via G. Falloppio 50, 35121 Padua, Italy; 5Department of Experimental Medicine, University of Campania “Luigi Vanvitelli”, Via Luciano Armanni 5, 80138 Naples, Italy; alessandro.feola@unicampania.it; 6Department of Occupational Medicine, University of Rome “Tor Vergata”, 00133 Rome, Italy; cristiana.ferrari@ptvonline.it (C.F.); luca.coppeta@ptvonline.it (L.C.)

**Keywords:** COVID-19 vaccine, public health ethics, healthcare workers, nurse

## Abstract

Background: While the COVID-19 pandemic has spread globally, health systems are overwhelmed by both direct and indirect mortality from other treatable conditions. COVID-19 vaccination was crucial to preventing and eliminating the disease, so vaccine development for COVID-19 was fast-tracked worldwide. Despite the fact that vaccination is commonly recognized as the most effective approach, according to the World Health Organization (WHO), vaccine hesitancy is a global health issue. Methods: We conducted a cross-sectional online survey of nurses in four different regions in Italy between 20 and 28 December 2020 to obtain data on the acceptance of the upcoming COVID-19 vaccination in order to plan specific interventions to increase the rate of vaccine coverage. Results: A total of 531 out of the 5000 nurses invited completed the online questionnaire. Most of the nurses enrolled in the study (73.4%) were female. Among the nurses, 91.5% intended to accept vaccination, whereas 2.3% were opposed and 6.2% were undecided. Female sex and confidence in vaccine efficacy represent the main predictors of vaccine intention among the study population using a logistic regression model, while other factors including vaccine safety concerns (side effects) were non-significant. Conclusions: Despite the availability of a safe and effective vaccine, intention to be vaccinated was suboptimal among nurses in our sample. We also found a significant number of people undecided as to whether to accept the vaccine. Contrary to expectations, concerns about the safety of the vaccine were not found to affect the acceptance rate; nurses’ perception of vaccine efficacy and female sex were the main influencing factors on attitudes toward vaccination in our sample. Since the success of the COVID-19 immunization plan depends on the uptake rate, these findings are of great interest for public health policies. Interventions aimed at increasing employee awareness of vaccination efficacy should be promoted among nurses in order to increase the number of vaccinated people.

## 1. Introduction

While the COVID-19 pandemic has spread globally, health systems are overwhelmed by both direct and indirect mortality from other treatable conditions [1,2,3]. For a time, no reliable treatment existed for COVID-19, and the only effective measure available to control the spread of the virus was to reduce the frequency of close contact between people [4]. Social distancing saves lives but imposes enormous costs on society due to reduced economic activity [5]. Vaccination against COVID-19 was crucial to preventing and eliminating the disease, and so vaccine development for COVID-19 was fast-tracked worldwide [6]. Since December 2020, three vaccines for the prevention of coronavirus disease 2019 (COVID-19) have been authorized for emergency use by both the Food and Drug Administration (FDA) and by the European Medicines Agency (EMA) [7,8].

Currently, in Italy, demand for COVID-19 vaccines is expected to exceed supply during the first phase of the national immunization program, so the Ministry of Health has advised the Health Service regarding priority population groups for vaccination [9]. During the pandemic, healthcare staff were recognized as a major component of pandemic preparedness [10]. Therefore, the Italian government’s plan stated that healthcare staff were to be offered COVID-19 vaccination first. Over the last decade, however, vaccine uptake by healthcare personnel has been low [11], causing uncertainties regarding whether said personnel would accept COVID-19 vaccination in correlation with previous vaccine habits and other variables of an occupational or personal nature. The success of an immunization plan is derived not only from high vaccine efficacy but also from adequate vaccine uptake among the target population [12]. Therefore, in order to prevent the spread of COVID-19, measures to increase the recognition of COVID-19 vaccines are also critical. Factors that may influence the acceptance of COVID-19 vaccination need to be identified, especially among health care workers at high risk of SARS-Cov-2 infection, in order to implement targeted training interventions [13].

Despite the fact that vaccination is commonly recognized as the most effective approach to preventing infection and reducing mortality from infectious diseases, vaccine hesitancy, acceptance delay, and vaccine refusal are growing phenomena among the general population and healthcare professionals [14,15].

According to the World Health Organization (WHO), vaccine hesitancy is a global health issue. The main factors contributing to hesitancy are a lack of confidence in vaccines (and fear of the potential hazards, including misconceptions about the risk of infection following vaccination), poor understanding of the need to vaccinate (e.g., underestimation of disease severity) or of the value of the vaccine, and difficulties accessing the vaccine [16]. Recent findings show that mandatory vaccination policies and strategies that improve vaccine accessibility are likely to be effective, whereas education policies are often ineffective [17,18].

Vaccination hesitancy in healthcare workers in Italy is increasing [19], thus their acceptance of the COVID-19 vaccination cannot be presumed, which will impact the current pandemic response.

Several studies have explored variables related to or affected by the uptake of vaccines among healthcare employees. A systematic review [20] of the variables that affect healthcare workers’ attitudes towards vaccination for influenza showed a higher probability of vaccine acceptance if the vaccine is perceived to be safe and effective and the infection severe. Certain studies have documented that operating in a high-risk environment with suspected or confirmed patients can be a determining factor for vaccination uptake [21]. These factors should be given greater emphasis during the COVID-19 pandemic, as healthcare workers’ perceptions of infection risk and disease severity may change when exposed to high-risk environments with COVID-19 as opposed to other pathogens.

## 2. Materials and Methods

We conducted a cross-sectional online survey on nurses in four different regions in Italy between 20 and 28 December 2020 to obtain data on the acceptance of COVID-19 vaccination in order to plan specific interventions to increase the rate of vaccine coverage.

The objective of the questionnaire was to collect information in order to assess the attitude of nurses towards vaccination, immediately prior to the start of large-scale vaccination. The information was collected anonymously, for statistical purposes only, for study and research in the field of vaccination prevention by the EEHTA research center of the CEIS of the University of Rome Tor Vergata, the chair of Occupational Medicine of the Department of Medicine and Health Sciences of the University of Molise, and the chair of Occupational Medicine of the University of Rome Tor Vergata.

We used a modified version of a validated Italian Institute of Health’s questionnaire regarding the psychological impact of COVID-19 infection among the Italian population [22].

The questionnaire was administered via LimeSurvey© (2006-2021 LimeSurvey GmbH, Hamburg, Germany), a platform which facilitates online surveys and questionnaires. The platform enables the use of different types of questions with the option to insert conditions and hierarchical dependencies between the questions themselves. The surveys, once created, are activated and distributed by means of the questionnaire’s unique link, which can be public or with restricted access.

In this specific case, the questionnaire was public and circulated using mailing lists of nursing staff primarily from the facilities conducting the study and subsequently using the mailing lists of nursing staff made available through presidents of the degree courses in Nursing Science at the aforementioned universities. This distribution enabled the swift collection of a total of 531 questionnaires, which were subsequently subjected to statistical and descriptive analyses.

The survey investigated four different areas: demographic and work-related factors; the socio-economic impact of COVID-19 on the operator’s situation; concern about infection; and vaccine-related behavior and intentions.

Participants were asked to indicate (1) whether they had accepted or intended to accept COVID-19 vaccination (“intend/not intend to accept” or “undecided”) and (2) whether they had accepted other occupational vaccines (“accepted”, “refused” or “accepted some and refused others”). Aside from their intention to be vaccinated, participants were also asked whether they had concerns regarding the efficacy and safety of the COVID-19 vaccine.

Regarding work-related and individual factors, we collected employment data (high-risk or non-high-risk setting, public or private service) and using a five-point Likert scale, both on perceptions as to the probability of their being infected with SARS-Cov-2 (“highly unlikely”, “unlikely”, “possible”, “likely” and “highly likely”) and on perceptions as to the potential severity of COVID-19 if infected (“not severe”, “probably not severe”, “possibly severe”, “probably severe” and “very severe”).

In addition, we collected data regarding the impact of the COVID-19 pandemic on the socio-economic situations (reduced earnings, increased time spent at work, increased workload) of the participants and their relatives.

All data were processed using Stata software, version 11.0 (Statacorp LP, 4905 Lakeway Drive, College Station, TX, USA)

The data on demographics (age class, gender), work-related factors, the socio-economic impact of COVID-19 on the operators’ situations, and concerns regarding the likelihood of infection, concerns about the severity of infection, and on vaccination acceptance and intention were reported as descriptive statistics.

To analyze the determinants of COVID-19 vaccine acceptance, we created a two-way cross-table between all collected variables, including previous acceptance of recommended vaccines and intention/acceptance of the COVID-19 vaccine.

We performed a multinomial logistic regression analysis to explore the association between attitudes to COVID-19 vaccination and work-related factors, the socio-economic impact of COVID-19 and the perceived likelihood and severity of infection after adjusting for age and gender. (Raosoft software Raosoft, Inc. 6645 NE Windermere Road, Seattle, WA, USA) was used for calculating the sample size. Keeping the margin of error at 5%, confidence interval at 95% and a population size of 5000, the sample size was calculated as 357.

Due to the small sample size, we grouped “highly unlikely”, “unlikely” and “possibly” into one category, “unlikely”, and “likely” and “highly likely” into just “likely” in the multiple regression analysis. We also regrouped the perceptions as to the potential severity of COVID-19 if infected as follows: “not severe”, “probably not severe” and “possibly severe” were replaced with “not severe”, and “probably severe” and “very severe” were recoded as “severe”.

Separately, we evaluated the difference between reasons for refusing and reasons for indecision on COVID-19 vaccination acceptance using Chi-square test.

## 3. Results

We estimated that the potential number of the subjects included in the mailing list for the survey was about 5000, but the rate of nurses who effectively opened the link of the LimeSurvey© platform is unknown. According to some published studies, the average reach is between 5 and 6% [23], but we can suppose a larger percentage due to the high HCW interest about COVID-19 vaccine.

A total of 531 nurses completed the online questionnaire.

Among the nurses enrolled in the study, 73.4% were female (Table 1). The majority of the study population were between 46 and 67 years of age (288/531; 54.2%), whereas the 18–25 and 26–45 age classes were less represented (7.3% nurses aged 18–25, 37.9% aged 26–45, 0.6% aged above 67).

Regarding occupational variables, we found that 66.7% worked in hospital settings, whereas 33.3% of nurses were employed in other settings. Most (72.3%) worked in high-risk settings (“high” or “very high” probability to meet a COVID-19 patient). Moreover, 84.7% had had a relative or friend affected since the start of the pandemic, and 7.3% reported bereavement due to COVID-19.

Regarding the subjective perception of infection risk, 59.9% of the participants reported anxiety “often” or “almost always” throughout the day; 68.4% felt that their likelihood of becoming ill was “high” or “very high”, and 18.6% felt that, if infected, their prognosis was “severe” or “very severe/death”. With respect to vaccine efficacy, 62.7% believe it is “effective” or “very effective” in protecting them from infection; 6.8% believed it is “partially effective” while 0.6% believed that the vaccine is ineffective. In addition, 15.8% of the study were either “concerned” or “very concerned” about the possible side effects of vaccination. Overall, 91.5% said they intended to accept vaccination, 2.3% were opposed to vaccination, and 6.2% were undecided. The main results are shown in Table 1.

The association between intention to be vaccinated and the main variables investigated by means of the questionnaire was analyzed by means of a logistic regression model (Table 2). The dependent variable of the model was vaccination intention. This was derived from the question designed to elicit participant propensity to be vaccinated against COVID-19. Responses were recoded to create a dichotomous variable that assumed a value of 1 if the participant had a definite intention to be vaccinated (answer “Yes” to the question), otherwise it assumed a value of 0 (answer “No” or “Don’t know”).

All other questions in the questionnaire were used as independent variables, that is, as regressors to explain the target variable.

The logistic regression model was estimated through SPSS software, release 26.0 (IBM Corporation, Armonk, NY, USA) by using the backward option, which eliminates, one by one, all independent variables that are not statistically significant in determining the probability of observing the event, and all variables the information from which is already explained by other variables. Finally, we found that the variables selected by the model as predictors of the probability of the intention to be vaccinated were sex (male or female) and confidence in vaccine efficacy in terms of protection from infection (“high” or “very high” vs. “not at all” and “low-very low”).

## 4. Discussion

We conducted this study in December during the epidemic, immediately before the vaccination campaign began. In Italy, since the beginning of the epidemic, more than 130,000 cases have been reported among healthcare workers, most of them nurses, in addition to numerous deaths [24,25].

To our knowledge, this is the first study performed in Italy during the period (December 2020) immediately preceding vaccination, whereas the results of the vaccination campaign in other countries were already available.

Despite the epidemiological situation, the level of COVID-19 vaccine intention found in the study population was suboptimal, in addition to a significant number of those being undecided.

The effectiveness of an immunization plan against an infectious disease depends on both vaccine efficacy and the vaccine uptake rate; low vaccination coverage can seriously affect the efficacy of the vaccination plan by reducing or eliminating the herd effect [26].

Although the percentage of participants willing to be vaccinated in our survey appears to be high, the presence of a non-insignificant number of nurses who were opposed or undecided can compromise hospital health policies and put the safety of fragile patients with whom they come into contact at risk. In previous studies regarding post-exposure screening for COVID-19, it was found that contact between healthcare workers is the main source of intrahospital infection rather than contact through assisting hospitalized patients [27]. Although there have been heated debates surrounding the potential compulsory use of the COVID-19 vaccine among healthcare workers, no real decision has been taken by the Italian government to date.

With respect to the intention to be vaccinated for COVID-19, we found a good indicator in the participants’ influenza vaccination history. Related findings were observed in previous research that prior vaccine approval was closely related to seasonal influenza vaccination acceptance and H1N1 vaccination [28].

Regarding intention to be vaccinated, we found good correlation between the acceptance or refusal of the COVID-19 vaccine and vaccination for other biological agents including influenza.

Similar findings were found in previous studies that showed a strong link between the acceptance of seasonal influenza vaccination and H1N1 vaccination [29]. Vaccination acceptance or refusal can be considered an individual habit [30] that can be applied to various vaccines for diseases with similar transmission routes and characteristics. However, vaccination approval rates for COVID-19 well above those shown for influenza both historically and during the last season can, in our opinion, be explained by the significant impact of the COVID-19 pandemic on health systems, the consequent global expectations surrounding the vaccine, and also by the emotional impact linked to the severity of the clinical situations encountered by operators when providing assistance in inpatient and intensive care unit settings. This explanation is, in our opinion, supported by the statistical correlation found between the presence of bereavement among nurses’ friends and family members and intention to be vaccinated.

Female sex, as expected, was positively related to the intention to be vaccinated; this observation is consistent with previous findings [17,19,31,32].

Based on a health belief model, having received a previous diagnosis of COVID-19 would have been expected to be a significant effector for the vaccine acceptance, but it was not. These subjects could be more sensitized to the risk of infection, though on the contrary, they could still consider themselves protected and postpone vaccination, especially in the case of a recent infection. As it has been very much emphasized by the scientific literature and institutions that a previous infection is not necessarily associated with lasting protection, this could have determined a more cautious attitude of these subjects.

Contrary to expectations, concerns with vaccine safety did not affect the acceptance rate in our population, whereas the results of our study showed a strong, statistically significant link between attitude toward vaccination and belief in vaccine efficacy. This factor appears to be a key determinant of vaccination intention despite other predictors (such as gender and age).

In a large survey carried out in France on 1554 healthcare operators, the authors found that COVID-19 vaccine acceptance was at 76.9% with a statistical relation to older age, male gender, influenza vaccination and individual perception of risk for COVID-19. Nurses were also less likely to accept vaccination than physicians [33]. The acceptance rate was substantially lower than that demonstrated in our study, but it should be considered that the study was conducted at a time when COVID-19 vaccination was still in the experimental phase, infection rates among the population were declining, and the second wave of infections had not yet occurred.

In a study from Hong Kong, less than two-thirds of the 1205 nurses who participated intended to accept the COVID-19 vaccine when it became available. The authors suggested that the decline in work-related stress among nurses that occurred in the post-pandemic period may have lowered the nurses’ inclination to accept the COVID-19 vaccine [34].

A study involving 735 students in Italy showed that 86.1% of them would choose to be vaccinated against COVID-19; the average age of the study population was very different from our sample population and this may have contributed to a different level of awareness [35].

In a survey conducted on 624 people living in Italy, 75.8% of them intended to accept the vaccine. Their decision was not linked to worry or institutional trust but was statistically related to beliefs about the non-natural origin of the virus [36].

Our findings suggest that future governmental strategies to promote COVID-19 vaccination should focus on vaccine efficacy and safety, for example by highlighting the decrease in the incidence rate for disease and hospitalization among vaccinated workers as compared with others. The serologic assessment of antibody titer, although not currently recommended by the CDC on the basis of the lack of a recognized protective level, could nevertheless prove useful in providing immediate confirmation of vaccine efficacy to workers and colleagues through the evidence of any type of antibody response.

Nurses from the private sector should be included first in the free COVID-19 vaccination program as they expressed a high vaccine acceptance rate, especially since a large number of all outpatient services in Italy are likely to be the first diagnosis point for patients. Moreover, considering the low willingness to be vaccinated among those who find vaccination for both COVID-19 and other pathogens unnecessary, policies to raise awareness of vaccine efficacy may also prove useful in attempting to increase immunization rates for major occupational pathogens and seasonal influenza. Moreover, since previous studies on acceptance rates for seasonal influenza vaccination and serological studies on immunity rates for occupational infectious diseases showed unacceptable levels of protection in Italy, awareness activities for vaccination may result in lasting benefits in global preventive health policies.

One possible limitation of our work is that, due to the study design, the survey was randomly sampled through a list of workers. This fact should have resulted in a sample selection bias and in the limited generalizability of the results. In fact, subjects opposed to vaccination may have been more reluctant to respond than those intending to be vaccinated. The complete anonymization of the questionnaire and sending it through a professional association should have allowed to control this possible confounding factors. Moreover, the statistical power of the study was limited and the real response rate among nurses is unknown. However, according to data from the literature that estimate the average reach to be between 5 and 6% [23], we can suppose that a large percentage of subjects who opened the link of the survey fulfilled the questionnaire. Regarding the sample size, the need to complete data collection among nurses in the short time period between the announcement of the vaccination campaign and its beginning (about one month) is the reason for the relatively limited number of subjects enrolled in the study.

We did not investigate whether the occupational physicians of operators’ facilities were involved in the promotion of vaccine acceptance among those subjects. It is a crucial question since vaccination attitude in HCWs may be widely affected by occupational physicians’ incomplete knowledge of vaccine recommendations. Based on the legal nature of the employment relationship, we can speculate that the educational contribution of the competent physician may have been relevant—especially for nurses working in hospitals [37].

## 5. Conclusions

Despite the epidemiological situation and the availability of a safe and effective vaccine, vaccine intention for COVID-19 among nurses is suboptimal, and the percentage of those undecided is significant.

The study was performed in the very early stages of the Italian vaccination campaign; therefore, when nurses were asked to participate, most of them were uncertain about the time and setting of eventual vaccination, and such a factor may have influenced their actual acceptance of vaccination.

Perceived vaccine efficacy was the main determinant of vaccine intention among the study population. Future strategies to promote COVID-19 vaccination should focus on vaccine efficacy and safety. Serologic assessment of antibody response to vaccines may prove useful in providing the confirmation of immune response and in increasing the awareness of vaccine efficacy among hesitant nurses.

## Figures and Tables

**Table 1 vaccines-09-00500-t001:** Main results of questionnaire by intention to be vaccinated among study population.

		Intention to Be Vaccinated for COVID-19					
		No		Yes		Undecided	
		N	Row N %	N	Row N %	N	Row N %
Sex	Female	9	2.30	363	93.10	18	4.60
	Male	3	2.10	123	87.20	15	10.60
Age	18–25	0	0.00	39	100.00	0	0.00
	26–45	6	3.00	183	91.00	12	6.00
	46–67	3	1.00	264	91.70	21	7.30
	>67	3	100.00	0	0.00	0	0.00
Regions	Campania	3	2.00	129	87.80	15	10.20
	Lazio	6	4.10	129	87.80	12	8.20
	Molise	13	3.80	301	89.58	22	6.55
	Veneto	3	4.00	64	85.33	8	10.67
Working area	Other	3	4.30	63	91.30	3	4.30
	Outpatient clinic	3	12.50	18	75.00	3	12.50
	Home Assistance	0	0.00	15	100.00	0	0.00
	Hospital	3	0.80	327	92.40	24	6.80
	Nursing Home	0	0.00	3	100.00	0	0.00
	Public Health	3	4.50	60	90.90	3	4.50
Probability of assisting COVID-19 patients	High	3	2.30	123	93.20	6	4.50
	Very high	3	1.20	231	91.70	18	7.10
	Low	0	0.00	48	94.10	3	5.90
	Very low	0	0.00	27	90.00	3	10.00
	Medium	6	9.10	57	86.40	3	4.50
Family members, friends or colleagues affected	No	0	0.00	81	100.00	0	0.00
	Yes	12	2.70	405	90.00	33	7.30
Bereavement due to COVID-19	No	12	2.40	450	91.50	30	6.10
	Yes	0	0.00	36	92.30	3	7.70
Concerned about COVID-19 during the day	Never	3	20.00	9	60.00	3	20.00
	Hardly ever	6	15.40	33	84.60	0	0.00
	Almost always	0	0.00	90	100.00	0	0.00
	Often	3	1.30	210	92.10	15	6.60
	Sometimes	0	0.00	144	90.60	15	9.40
Likelihood of being infected with SARS-CoV-2 during activities	Likely	9	4.10	204	91.90	9	4.10
	Highly likely	0	0.00	129	91.50	12	8.50
	Unlikely	3	9.10	27	81.80	3	9.10
	Possibly	0	0.00	123	93.20	9	6.80
	Highly unlikely	0	0.00	3	100.00	0	0.00
Potential severity of COVID-19 in case of contagion	Severe	3	3.40	81	93.10	3	3.40
	Very severe/Death	0	0.00	12	100.00	0	0.00
	Probably not severe	3	2.90	96	91.40	6	5.70
	Probably severe	3	1.10	264	92.60	18	6.30
	Not severe	3	7.10	33	78.60	6	14.30
Confidence in vaccine efficacy	None	6	3.80	141	88.70	12	7.50
	Very high	0	0.00	111	100.00	0	0.00
	High	0	0.00	219	98.60	3	1.40
	Very low	3	100.00	0	0.00	0	0.00
	Low	3	8.30	15	41.70	18	50.00
Concerned about vaccine safety (side effects)	Partly	3	1.50	189	94.00	9	4.50
	Very much	3	14.30	12	57.10	6	28.60
	Much	6	9.50	39	61.90	18	28.60
	Not at all	0	0.00	90	100.00	0	0.00
	Little	0	0.00	156	100.00	0	0.00

**Table 2 vaccines-09-00500-t002:** Factors influencing participant vaccine intention (logistic regression analysis).

	O R	95% C.I.	*p*-Value
SARS-CoV-2 infection diagnosis	0.51	0.11–2.25	0.37
Positive family members, friends or colleagues	0.12	0.01–1.19	0.07
Mourning due to COVID-19	0.27	0.04–1.64	0.16
Concerned about COVID-19 during the day	3.92	0.66–23.28	0.13
Likelihood of contracting COVID-19 during work-related activities	2.96	0.71–12.25	0.13
Clinical evolution in case of contagion	1.24	0.19–7.73	0.82
Concerned about vaccine safety (side effects)	2.68	0.92–8.67	0.06
Influenza vaccine	20.82	1.12–385.75	<0.05
Female sex	0.13	0.03–0.55	<0.01
Confidence in vaccine efficacy	33.53	5.68–197.70	<0.01

## Data Availability

The data presented in this study are available by request from the corresponding author. The data are not publicly available for ethical reasons.

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
