# Peer review of "Intention to Be Vaccinated for COVID-19 among Italian Nurses during the Pandemic"

_vaccines, 2021, doi:10.3390/vaccines9050500_

Round 1

Reviewer 1 Report

I was invited to revise the paper entitled "Intention to Vaccinate For COVID-19 Among Italian Nurses During the Pandemic". The paper aimed to investigate the attitude towards COVID19 vaccination among italian nurses. The topic is interesting and can improve the knowledge in this field. In my knoledge it is the first paper in this setting performed in Italy.

I have some observation that need to be addressed:

  • Sample size estimation was lacking;
  • Validation of the questionnaire was missing;
  • In Table 2 Authors should present OR with 95%CI instead of B. In addition Sign should be replaced by p-value;
  • About discussion, Authors should also highlight age differences and gender differences in vaccine hesitancy among HCWs, as reported in previous studies (ex. 10.3390/vaccines8020248 and 10.1016/j.vaccine.2012.04.098)

    Author Response

    We want to thank the reviewers for their work that allowed us to improve the manuscript. We responded precisely to every input.

    1. Sample size estimation was lacking;

    A statement regarding power analysis was added in the methods section.

    1. Validation of the questionnaire was missing;

    We used modified version of a validate Italian Institute of Health’s questionnaire regarding the psychological impact of COVID-19 infection among the Italian population.

    1. In Table 2 Authors should present OR with 95%CI instead of B. In addition Sign should be replaced by p-value;

    We’ve modified the table 2.

    1. About discussion, Authors should also highlight age differences and gender differences in vaccine hesitancy among HCWs, as reported in previous studies (ex. 10.3390/vaccines8020248 and 10.1016/j.vaccine.2012.04.098)

          Discussion section was implemented according to reviewer’s suggestions.

    Reviewer 2 Report

    Estimated Authors,

    Estimated Editors,

    I've read with great interest this paper from the study group lead by Aurilio on the intention to being vaccinated against COVID-19 Among Italian Nurses During the Pandemic.

    In their study, Authors have collected knowledge, attitudes, and beliefs of a substantial sample of nurses from 4 Italian regions, and summarized potential effectors towards the outcome variable to accept SARS-CoV-2 vaccine. Eventually, quite general factors were identified in logistic regression as true effectors, i.e. "gender (Male or Female) and confidence in vaccine efficacy in 183 terms of protection from infection".

    In my opinion, this paper is quite interesting, but a series of improvements are required before a full publication on Vaccines, and namely:

    1. It is rather unclear whether a preventive power analysis was performed or not; please report in clear terms whether the study participants were recruited as a convenience sample or not.
    2. The reduced participating rate (around 10%) make possible a significant selection bias, that should be extensively discussed in the Discussion section. 
    3. As Italy is quite heterogenous in terms of vaccine practice and medical/scientific literacy, Authors should report in table 1 the actual composition of their study group (i.e. how many participants were recruited from Region 1, 2, etc...)
    4. It would be particularly interesting to know whether the occupational physician of the parent healthcare facilities were involved or not in the promotion of vaccine among healthcare workers, as occupational physicians are able to significantly improve vaccine acceptance among HCW (https://pubmed.ncbi.nlm.nih.gov/28661516/).
    5. According to the health belief model, having received a previous diagnosis of SARS-CoV-2 would have been expected to be a significant effector for the eventual acceptance of vaccine, while it wasn't. Please discuss in larger extent this topic.
    6. The study was performed in the very early stages of Italian vaccination campaign. It should be explained, in further details, that when Nurses were asked to participate, most of them were uncertain about time and settings for eventual vaccination, and such factor may have influenced their actual acceptance of vaccination.
    7. the term: "vaccine intention" may be somewhat misleading, as we are dealing with "acceptance of vaccination" of "to be vaccinated". Please reframe in order to avoid possible misunderstandings.

    Author Response

    We want to thank the reviewers for their work that allowed us to improve the manuscript. We responded precisely to every input.

    1. It is rather unclear whether a preventive power analysis was performed or not; please report in clear terms whether the study participants were recruited as a convenience sample or not.

                 A statement regarding power analysis was added in the methods section.

    1. The reduced participating rate (around 10%) make possible a significant selection bias, that should be extensively discussed in the Discussion section. 

    The possible selection bias due to low participating rate was discussed in the text.

    1. As Italy is quite heterogenous in terms of vaccine practice and medical/scientific literacy, Authors should report in table 1 the actual composition of their study group (i.e. how many participants were recruited from Region 1, 2, etc...)

    Data regarding region composition and respective outcomes were showed in table 1.

    1. It would be particularly interesting to know whether the occupational physician of the parent healthcare facilities were involved or not in the promotion of vaccine among healthcare workers, as occupational physicians are able to significantly improve vaccine acceptance among HCW (https://pubmed.ncbi.nlm.nih.gov/28661516/).

          The role of occupational physicians in improving vaccine acceptance was discussed in the       

           appropriate section.

    1. According to the health belief model, having received a previous diagnosis of SARS-CoV-2 would have been expected to be a significant effector for the eventual acceptance of vaccine, while it wasn't. Please discuss in larger extent this topic.

          We discussed in larger extent this topic in the appropriate section.

    1. The study was performed in the very early stages of Italian vaccination campaign. It should be explained, in further details, that when Nurses were asked to participate, most of them were uncertain about time and settings for eventual vaccination, and such factor may have influenced their actual acceptance of vaccination.

          Conclusions section was implemented according to reviewer’s suggestions.

    1. the term: "vaccine intention" may be somewhat misleading, as we are dealing with "acceptance of vaccination" of "to be vaccinated". Please reframe in order to avoid possible misunderstandings.

           We reframed the text, according to reviewer’s suggestions.

    Reviewer 3 Report

    This article describes the results of a survey of nursing staff in three regions of Italy regarding their willingness to receive a coronavirus vaccine. Unfortunately it is marred by its sample selection method and the resulting analysis suffers from this. In particular:

    1. The survey was not randomly sampled, but purposively sampled through lists of nurses
    2. It only had a 10% response rate, which is insufficient to draw any conclusions of any generalizability

    As a result of the low response rate (only 531 nurses) means that there is no power in the analysis. We can see this from the logistic regression results, where the very small number of vaccine-abstainers leads to very wide confidence intervals in estimates. For example (Table 2), the 95% CI for the effect of past influenza vaccine ranges from 1.12 to 385.75. This is simply too wide for a realistic study.

    Given these problems, I cannot recommend publication of this article.

    Author Response

    we want to thank the reviewer for his work and comments that have allowed us to improve our work.
    We want to respond to comments with some considerations.
    1. The survey was not randomly sampled, but purposively sampled through lists of nurses.
    The work focuses on the role of nurses. The aim is, among other things, to know the intention regarding the vaccine of this specific group as it is directly involved in the care and assistance of the sick during the pandemic even in particular contexts such as nursing homes for the elderly. For this reason we had to necessarily select the sample on the basis of the profession and not open it to a general sample of the population. For this reason we have used an IT platform. We have tried to better explain this point on p. 4 of the article in order to clarify it better. Furthermore possible confounding factor were discussed in the appropriate section.
    2. It only had a 10% response rate, which is insufficient to draw any conclusions of any generalizability
    Thanks for this comment. In fact the 10% response rate was wrongly reported in the text. Statistical power of the study was limited and the real response rate among nurses is unknown. However, according to data from literature that estimate the average reach to be between 5–6%, we can suppose that a large percentage of subjects who opened the link of the survey fulfilled the questionnaire. A statement regarding the sample size were discussed in the limit section. It is also necessary to consider that we carried out the study at a particular time in history during the pandemic and in a very short period of a few days in order to make the results public before the start of the vaccination campaign in Italy.
    Ultimately, we believe our study is worthy of publication as it takes into consideration a peculiar aspect inherent in the vaccination intention of healthcare professionals who have fought the pandemic in one of the most affected countries in the world. We think this study may be of some interest to Vaccine readers as indicated by other reviewers.

    Round 2

    Reviewer 1 Report

    The paper in now acceptable for publication

    Author Response

    thanks for the work and advice from the reviewer

    Reviewer 3 Report

    Thank you for clarifying the response rate issue. Nonetheless, my opinion remains unchanged: the response rate is simply too low to enable any assessment of the validity and generalizability of the results.